# Reconfiguration of Gut Microbiota and Reprogramming of Liver Metabolism with Phycobiliproteins Bioactive Peptides to Rehabilitate Obese Rats

**DOI:** 10.3390/nu14173635

**Published:** 2022-09-02

**Authors:** Jing Liu, Dongyu Zhen, Changbao Hu, Yawen Liu, Xuanri Shen, Pengcheng Fu, Yanfu He

**Affiliations:** 1State Key Laboratory of Marine Resource Utilization in South China Sea, Hainan University, Haikou 570228, China; 2School of Food Science and Engineering, Hainan University, Haikou 570228, China; 3Hainan Provincial Engineering Research Centre of Aquatic Resources Efficient Utilization in the South China Sea, Hainan University, Haikou 570228, China; 4Weihai UIC Biotechnology, Inc., 23 Shenzhen Road, Gaocun, Wendeng District, Weihai 264408, China

**Keywords:** phycobiliproteins, bioactive peptides extract, obesity, gut microbiome, serum metabolome

## Abstract

Phycobiliproteins (derived from *Arthrospira platensis*) bioactive peptide extracts (PPE) possess multiple pharmacological effects in the mitigation of human metabolic disorders. The role of PPE in the treatment of diet-induced obesity and the understanding of the underlying mechanism between the gut microbiome and metabolic blood circulation for obese patients remains poorly understood. In this study, we showed that PPE attenuated obesity by reducing body weight, and ameliorated glucose and lipid indexes in serum. In particular, PPE is postulated to mitigate liver steatosis and insulin resistance. On the other hand, dietary treatment with PPE was found to “reconfigure” the gut microbiota in the way that the abundances were elevated for *Akkermansia_muciniphila*, beneficial *Lactobacillus* and *Romboutsia*, SCFA-producing species *Faecalibacterium prausnitzii*, Lachnospiraceae_bacterium, Clostridiales_bacterium, probiotics *Clostridium* sp., *Enterococcus faecium,* and *Lactobacillus_johnsonii*, while the abundance of Firmicutes was reduced and that of Bacteroidetes was increased to reverse the imbalance of Firmicutes/Bacteroidetes ratio. Finally, the metabolomics of circulating serum using UHPLC-MS/MS illustrated that PPE supplementation indeed promoted lipid metabolism in obese rats. As summary, it was seen that PPE reprogrammed the cell metabolism to prevent the aggravation of obesity. Our findings strongly support that PPE can be regarded as a potential therapeutic dietary supplement for obesity.

## 1. Introduction

Obesity causes metabolic disorders which may lead to a series of diseases, such as diabetes, hypertension, cardiovascular and cerebrovascular diseases, etc., which seriously affect people’s lives [1]. Genetic, social environmental, psychological and many other factors are related to the prevalence of obesity, while the more direct cause and effect of overweight and obesity are that the energy intake consistently exceeds the energy expenditure of the patients [2]. The gut microbiota in human gastrointestinal tracts plays an important role in the process of energy intake, transformation, and storage [3,4]. Intriguingly, current evidence suggests that gut microbiota reprogramming may alleviate metabolic diseases such as obesity, non-alcoholic fatty liver disease, and type II diabetes [5,6,7,8,9]. Proteins play an extremely important role in maintaining the metabolism and transport of various substances, maintaining fluid and acid-base balance, and antibody synthesis [10]. Many observational and interventional evidence confirmed that increasing protein intake, especially plant-based proteins has a positive effect on modifying metabolic disease [10,11,12,13,14]. Peptides from natural sources possess different biological activities, mostly concerned with the treatment of human diseases [15,16,17,18].

*Arthrospira platensis* is a novel and sustainable source of dietary proteins, constituting 60–70% of proteins of its dry weight, it also contains a variety of essential amino acids, vitamins, and minerals [19]. Phycobiliproteins derived from *Arthrospira platensis* were brightly colored, highly fluorescent [20], and nutritive with nutraceutical properties: such as hypolipidemic [21], hypoglycemic [22], antihypertensive [23], anti-inflammatory [24], and antioxidant activities [25]. Moreover, there are reports on the clinical tests and animal models in the literature that suggested phycobiliproteins bioactive peptides have the potential to regulate blood glucose and lipid levels [26]. In particular, phycobiliproteins peptides are seen to be able to activate the insulin signaling pathway [27] and improve lipid metabolism disorders [12]. Yet, the intrinsic effect of PPE on gut microbiota and the underlying mechanism of modulating glucose-lipid metabolism in obesity remains poorly understood.

Obesity is a fast-evolving research field in which multidisciplinary efforts are needed to understand the mechanisms of obesity and to seek an effective approach to mitigate it. Based on previous studies in the literature and our own preliminary work, we hypothesize that there exists possible crosstalk among gut, liver, and circulating metabolites and bioactive peptides are able to alter the composition of gut microbiota and liver glucose-lipid metabolism in obese rats. To examine this hypothesis in this study, we assessed the alteration of the rat intestinal metagenome and serum metabolome in response to PPE supplementation for obese rats and elucidate the glucose-lipid metabolic pathway regulation in the liver. As summary, we established a metabolic model to describe how PPE stimulates the PI3K-Akt signaling pathway and glycogen synthesis, regulates gluconeogenesis, and manipulates fatty synthesis in the liver. The results from this study may contribute to understanding the role of PPE supplementation in the relief of obesity and glucose-lipid disorder-related syndromes, focusing on the interplay among gut microbiota and circulating metabolites in diet-induced obese rats.

## 2. Materials and Methods

### 2.1. PPE

*Arthrospira platensis* phycobiliprotein was donated by Xindaze Biotech Co., Ltd. (Fuqing, Fujian, China). PPE was prepared by the enzymatic hydrolysis of phycobiliprotein according to the method of our previous study [28]. Briefly, food-grade dried phycobiliprotein was mixed with phosphate buffer (pH 7.1). Then at 55 °C, added bromelain protease (200 units/mg) purchased from Yuan Ye Co., Ltd. (Shanghai, China). The mixture was put in a water bath shaker for 6 h. After that, the hydrolysate was centrifuged at 8000× *g* for 15 min using a bench centrifuge (Eppendorf AG, Saxony, Germany), and then the bioactive peptides were obtained through passing ultrafiltration membranes (with a molecular weight (MW) cut-off of 3 kDa). All of the recovered peptides were lyophilized for further study.

### 2.2. Rats and Diets

Six-week-old male Sprague Dawley (SD) rats (weighing 200 ± 20 g) were obtained from Hunan SJA Laboratory Animal Co. Ltd. (Changsha, China; Certificate number: SCXK (Xiang) 2019-0004). On arrival, all the rats were randomly housed in individually ventilated-plastic cages (6 rats per cage) with a 12/12 h light/dark cycle, controlled temperature at 23 (±1) °C, 45 ± 5% humidity, and free access to water and standard full-value chow (Hunan SJA Laboratory Animal Co. Ltd., Changsha, China). All animals were handled according to the guidelines of the principle of Laboratory Animal Care of China Animal Health and Epidemiology Center (www.cahec.cn, accessed on 1 May 2021). All animal operations were approved by the Hainan University Institutional Animal Welfare and Ethical Committee (HNDX2021075).

After one week of acclimatization, the rats were randomly divided into two groups, including the high-fat diet group (*n* = 24, fed with high-fat diet H10045) and the chow diet group (Chow control, *n* = 6, fed with chow diet H10010). Two DIO series diets purchased from Vital River Laboratory Animal Co., Ltd. (Beijing, China) were used during the following 10 weeks experiment: high-fat diet H10045, composed of 45% fat, 20% protein, and 35% carbohydrate; chow diet H10010, composed of 10% fat, 20% protein, and 70% carbohydrate. After six weeks, the high-fat diet group of obese rats were randomly divided into four groups for different doses of PPE intragastric operation. Four groups included the high-dose PPE treatment group (PPE_H, *n*= 6, 500 mg/(Kg·BW·d)), the medium-dose PPE treatment group (PPE_M, *n* = 6, 250 mg/(Kg·BW·d)), the low-dose PPE treatment group (PPE_L, *n* = 6, 125 mg/(Kg·BW·d)), and the high fat diet group without PPE treatment (HFD control, *n* = 6, negative control group and treated with the same dose of water as PPE). We have designed the PPE doses for rats based on the guidance described in “Dose Design Method for Experimental Animal Pharmacology Tests” (in Chinese, http://www.lascn.net/Item/13064.aspx, accessed on 3 May 2021), in which the dose of animal experiment in vivo was increased by 2 or 3.16-fold when exploring the optimal dose. The PPE doses were determined in the following steps: (I) The oral PPE doses for healthy people were 2–10 g/d [29]. (II) It was converted into equivalent rats’ dose in terms of body surface area ratios between humans and rats. PPE doses were thus 210–1050 mg/(Kg·BW·d). A total of 250 mg/(Kg·BW·d) was selected as the medium dose. (III) There exists a rule for 2-fold differences among low, medium, and high doses [7,30,31]. We thus selected 2-fold increments for the three doses as shown below: high dose: 500 mg/(Kg·BW·d), medium dose: 250 mg/(Kg·BW·d), and low dose: 125 mg/(Kg·BW·d). The experimental schedule and grouping information are shown in Appendix A. Rats received allocated treatment once a day for four weeks by intragastric administration. The body weight was measured weekly. At the completion of the study, the rats were fasted for 12 h before sacrifice. The blood was collected and allowed to clot at room temperature for 15 min, then centrifuged (4 °C, 4000× *g*, 10 min) and the resultant serum was stored at −80 °C for subsequent analysis. The liver was harvested and part of it was immediately fixed in formalin for hematoxylin-eosin staining (HE) while the rest was immediately frozen in liquid nitrogen and stored at −80 °C for subsequent analysis. All the tissues were washed with precooled phosphate buffer saline (PBS) buffering before harvesting. Colon contents were collected and stored at −80 °C for subsequent analysis.

### 2.3. Glucose Tolerance Test

After four weeks of PPE treatment, glucose tolerance tests were conducted with a glucose solution (2 g/kg BW) by intragastric administration after 8 h fasting, blood glucose levels (mmol/L) were measured from the tail vein using a touch glucometer (Roche Diagnostics, Mannheim, Germany) at 0, 30, 60, 90, 120, 150, and 180 min after injection.

### 2.4. Serum Biochemical Analysis

Serum triglyceride (TG), total cholesterol (TC), low-density lipoprotein cholesterol (LDL-C), and high-density lipoprotein cholesterol (HDL-C) were estimated by corresponding reagent kits (Jiancheng, Nanjing, China) and used strictly according to the kit instructions. Adiponectin, leptin, insulin, tumor necrosis factor-α (TNF-α), interleukin 6 (IL-6), and interleukin 10 (IL-10) levels were tested by ELISA kits (Multisciences Biotech, Co., Ltd., Hangzhou, China).

### 2.5. Gene Expression

Total RNA was extracted from liver tissues with RNAiso Plus reagent (Takara, cat#9108, Beijing, China) according to the manufacturer’s instructions. The quality and concentration of total RNA were measured by a spectrophotometer (NanoDrop 2000, NanoDrop Technologies, LLC, Wilmington, USA). A total of 200 ng/uL total RNA was reversely transcribed using a HiScript III All-in-one RT SuperMix Perfect for qPCR Kit (Vazyme, Nanjing, China). Real-time qPCR was performed using a ChamQ Universal SYBR qPCR Master Mix kit (Vazyme, Nanjing, China) and ABI QuantStudio™ 6 Flex Real-time PCR System (ABI, CA, USA). The expression of target genes was normalized to the expression of Gapdh, and shown as fold change relative to the control group based on the 2^-ΔΔCt^ method. The primer sequences were shown in Appendix A.

### 2.6. Metagenomics

Total genomic DNA was extracted from colonic contents (the colon tissue was not scraped) using the FastDNA^®^ Spin Kit for Soil (MP Biomedicals, CA, USA) according to the manufacturer’s instructions. Concentration and purity of extracted DNA were determined with TBS-380 and NanoDrop2000, respectively. DNA extract was fragmented to an average size of about 400 bp using Covaris M220 (Gene Company Limited, Zhuhai, China) for paired-end library construction. Paired-end library was constructed using NEXTFLEX Rapid DNA-Seq (Bioo Scientific, Austin, TX, USA). Adapters containing the full complement of sequencing primer hybridization sites were ligated to the blunt end of fragments. Paired-end sequencing was performed on Illumina NovaSeq (Illumina Inc., San Diego, CA, USA) at Majorbio Bio-Pharm Technology Co., Ltd. (Shanghai, China) using NovaSeq Reagent Kits according to the manufacturer’s instructions (www.illumina.com, accessed on 10 October 2021).

The original sequencing data were evaluated by FastQC, and Trimmomatic filtering was applied to obtain relatively accurate and effective data. The paired-end Illumina reads were trimmed of adaptors, and low-quality reads (length < 50 bp or with a quality value < 20 or having N bases) were removed by fastp [32] (https://github.com/OpenGene/fastp, version 0.20.0, accessed on 12 October 2021). Reads were aligned to the rattus_norvegicus genome by BWA [33] (http://bio-bwa.sourceforge.net, version 0.7.9a, accessed on 12 October 2021) and any hit associated with the reads and their mated reads were removed. Metagenomic data were assembled using MEGAHIT [34] (https://github.com/voutcn/megahit, version 1.1.2, accessed on 12 October 2021), which makes use of succinct de Bruijn graphs. Contigs with a length being or over 300 bp were selected as the final assembling result, and then the contigs were used for further gene prediction and annotation.

Open reading frames (ORFs) from each assembled contig were predicted using MetaGene [35] (http://metagene.cb.k.u-tokyo.ac.jp/, accessed on 13 October 2021). The predicted ORFs with a length being or over 100 bp were retrieved and translated into amino acid sequences using the NCBI translation table (http://www.ncbi.nlm.nih.gov/Taxonomy/taxonomyhome.html/index.cgi?chapter=tgencodes#SG1, accessed on 14 October 2021). A non-redundant gene catalog was constructed using CD-HIT [36] (http://www.bioinformatics.org/cd-hit/, version 4.6.1, accessed on 14 October 2021) with 90% sequence identity and 90% coverage. Reads after quality control were mapped to the non-redundant gene catalog with 95% identity using SOAPaligner [37] (http://soap.genomics.org.cn/, version 2.21, accessed on 14 October 2021), and gene abundance in each sample was evaluated. Representative sequences of non-redundant gene catalog were aligned to the NCBI NR database with an e-value cutoff of 1 × 10^−5^ using Diamond [38] (http://www.diamondsearch.org/index.php, version 0.8.35, accessed on 14 October 2021) for species annotations. The microbiota analysis methods were that using the sum of the corresponding gene abundances to calculate the abundance of species, then the abundance of species in each sample was measured at the taxonomic level of Domain, Kingdom, Phylum, Class, Order, Family, Genus, and Species. Thus, the abundance profile at the corresponding taxonomic level can be constructed. The annotation was conducted using Diamond against the Kyoto Encyclopedia of Genes and Genomes (KEGG) database (http://www.genome.jp/keeg/, accessed on 18 October 2021) with an e-value cutoff of 1 × 10^−5^.

### 2.7. Metabolomic Study of Rat Serum by UHPLC-MS/MS

Metabolites were extracted from 100 µL rat serum sample using a 400 µL methanol: water (4:1, *v*/*v*) solution, with 0.02 mg/mL L-2-chlorophenylalanin as internal standard. The mixture was allowed to settle at −10 °C and treated by a high throughput tissue crusher Wonbio-96c (Shanghai wanbo biotechnology co., LTD) at 50 Hz for 6 min, then followed by ultrasound at 40 kHz for 30 min at 5 °C. The samples were placed at −20 °C for 30 min to precipitate proteins. After centrifugation at 13,000× *g* at 4 °C for 15 min, the supernatant was carefully transferred to sample vials for UHPLC-MS/MS analysis. As a part of the system conditioning and quality control process, a pooled quality control sample (QC) was prepared by mixing equal volumes of all samples. The QC samples were disposed of and tested in the same manner as the analytic samples. It helped to represent the whole sample set, which would be injected at regular intervals (every 10 samples) in order to monitor the stability of the analysis.

Chromatographic separation of the metabolites was performed on a Thermo UHPLC system equipped with an ACQUITY UPLC HSS T3 (100 mm × 2.1 mm i.d., 1.8 µm; Waters, Milford, USA). The mobile phases consisted of solvent A: 0.1% formic acid in water: acetonitrile (95:5, *v*/*v*) and solvent B: 0.1% formic acid in acetonitrile: isopropanol: water (47.5:47.5:5, *v*/*v*). The sample injection volume was 2 µL and the flow rate was set to 0.4 mL/min. The column temperature was maintained at 40 °C. The mass spectrometric data were collected using a Thermo UHPLC-Q Exactive HF-X Mass Spectrometer equipped with an electrospray ionization source operating in either positive or negative ion mode. The optimal conditions were set as followed: heater temperature, 425 °C; Capillary temperature, 325 °C; sheath gas flow rate, 50 arb; Aux gas flow rate, 13 arb; ion-spray voltage floating (ISVF), −3500 V in negative mode and 3500 V in positive mode, respectively; Normalized collision energy, 20–40–60 V rolling for MS/MS. Full MS resolution was 60,000, and MS/MS resolution was 7500. Data acquisition was performed with the Data Dependent Acquisition (DDA) mode. The detection was carried out over a mass range of 70–1050 *m*/*z*.

After UPLC-MS analyses, the raw data were imported into the Progenesis QI 2.3 (Nonlinear Dynamics, Waters, Milford, USA) for peak detection and alignment. The preprocessing results generated a data matrix that consisted of the retention time (RT), mass-to-charge ratio (*m*/*z*) values, and peak intensity. Metabolic features detected at least 80% in any set of samples were retained. After filtering, minimum metabolite values were imputed for specific samples in which the metabolite levels fell below the lower limit of quantitation and each metabolic feature was normalized by sum. A multivariate statistical analysis was performed using ropls (Version1.6.2, http://bioconductor.org/packages/release/bioc/html/ropls.html, accessed on 25 October 2021) R package from Bioconductor on Majorbio Cloud Platform (https://cloud.majorbio.com, accessed on 25 October 2021). Orthogonal partial least squares discriminate analysis (OPLS-DA) was used for statistical analysis to determine global metabolic changes between comparable groups. Variable importance in the projection (VIP) was calculated in the OPLS-DA model. *p* values were estimated with paired Student’s *t*-test on Single dimensional statistical analysis. Differential metabolites among two groups were summarized, and mapped into their biochemical pathways through metabolic enrichment and pathway analysis based on a database search (KEGG, http://www.genome.jp/kegg/, accessed on 30 October 2021).

### 2.8. Short-Chain Fatty Acids (SCFAs) Analysis in Feces

Fresh fecal samples were collected every week from each rat. Feces SCFAs were detected using a gas chromatography (GC) system as previously reported, with a few modifications [39]. Approximately 200 mg of the feces sample was weighed and added to 1 mL of water then homogenized (Biological Sample Preparation System, Life real co., LTD, China) for about 3 min. Then SCFAs were extracted by adding 0.15 mL 50% (*w*/*w*) H2SO4 and 1.6 mL of diethyl ether. After that, the samples were incubated in ice-cold water for 30 min and centrifuged at 5000× *g* for 15 min at 4 °C. The supernatant organic phase was analyzed by Agilent 6890 N gas chromatograph with a flame ionization detector (FID), according to the following program: 80 °C for 0.5 min, 80 °C raised to 150 °C at 4 °C /min, and then 150 °C increased to 230 °C at 20 °C /min held for 10 min. The injector temperature was 250 °C and the detector temperature was 270 °C.

### 2.9. Statistical Analyses

All the results were expressed as the mean and standard deviation. For multiple group comparison, a one-way analysis of variance (ANOVA) followed by Tukey’s post hoc test (SPSS26, Inc., Chicago, IL, USA) was performed to identify differences among means. Statistical significance is represented by * *p* < 0.05, ** *p* < 0.01, *** *p* < 0.001. Sample size and statistical tests are also indicated in the figure legends.

## 3. Results

### 3.1. PPE Reduced Body Weight and Ameliorated Serum Glucose and Lipid Indexes

In our previous study, PPE was demonstrated that it had several bioactivities including DPP-IV inhibitory activity, alpha-glucosidase inhibitory activity, ACE inhibitory activity, antioxidant and immunomodulatory activity by in silico and in vitro assessment [28]. Here, we further explored the bioactivities of PPE in obese rats. To investigate the efficacy of PPE on ameliorating obesity caused by high-fat diets, we firstly analyzed the indicators of growth performance and serum glucose and lipid indexes in rats. Five groups were set up: a negative control group induced by a high-fat diet, high, medium, and low dose PPE nutritional intervention groups, and a positive control group with a chow diet. After six weeks of high-fat diet, a significant difference was observed in body weight (obesity model group was 120.32% higher than the chow control group, *p* < 0.01), and increment TG contents (*p* < 0.05) indicated that the model of obesity was successfully constructed (Appendix A). The results showed that four weeks of PPE continuous nutritional intervention could significantly reduce the weight of obese rats (PPE_M group, *p* < 0.05) compared with the HFD group (Figure 1A). Furthermore, an oral glucose tolerance test revealed that more effective glucose clearance after glucose administration was also observed in the PPE treatment groups (Figure 1B). The serum insulin concentration of rats in the HFD control group (9.92 ng/mL, *p* < 0.05) was significantly higher than that of the Chow control group (9.11 ng/mL, Figure 1D), while the regulatory effect of PPE intervention was not obvious.

Regarding lipid metabolism, we observed that PPE-fed obesity rats had significantly reduced their levels of TG, TC, and LDL-C, while the levels of HDL-C were increased (Figure 1C). It was seen that PPE supplementation has not significantly improved the serum adiponectin level; however, serum leptin level was remarkably decreased (Figure 1E,F). It is well known that obesity would reduce adiponectin levels and increase leptin levels. Changes in these cytokines may trigger insulin resistance and cause fatty liver, as well as promote liver lesions and inflammation.

Our experimental results indicated that in the case of losing body weight and ameliorating serum glucose index, the PPE treatment group with medium-dose (250 mg/(Kg·BW·d)) showed a better effect than the other two groups. We did not observe the dose dependence in blood lipid level, however, high-dose of PPE was seen to significantly lower the leptin level in serum, which was in favor of mitigation of obesity. It is regretful that inflammatory cytokines levels in serum did not respond to treatment of PPE in a dose-dependent manner.

### 3.2. PPE Inhibited Liver Lesions and Inflammation in Obese Rats

To further investigate the functions of PPE on the liver of obese rats, pathological characteristics were examined. Liver steatosis and massive infiltration of inflammatory cells were observed in the HFD group, However, in the PPE-supplemented groups, inflammatory infiltration was found to disappear, and the excessive accumulation of lipid droplets was greatly alleviated, especially in PPE_M and PPE_H group (Figure 1G), which indicated that PPE intervention improved the obesity-induced liver disease.

After inflammatory cell infiltration was observed in liver tissue, serum inflammatory factors were tested (Figure 1H–J). It is noteworthy that serum TNF-α and IL-6 levels decreased significantly (*p* < 0.01), meanwhile serum IL-10 levels increased significantly (*p* < 0.05), which is indicative of the strong anti-inflammatory ability of PPE. The decrease of TNF-α and IL-6 can alleviate insulin resistance and liver steatosis caused by obesity.

### 3.3. PPE Attenuated Liver Steatosis and Insulin Resistance by Regulated Genes in Obese Rats’ Liver

Obesity is closely related to fatty liver which is formed by increased synthesis of fatty acids or decreased oxidation of fatty acids in hepatic tissue. In order to verify that the altered expression of lipid metabolism in obese rats is a direct result of PPE supplementation, we extracted RNA from liver tissue and performed qPCR on target mRNA. Importantly, the mRNA expression of *ACC* and *FASN* was markedly (*p* < 0.001) reduced by PPE feeding (Figure 2B,C). By stark contrast, they were significantly (*p* < 0.001) increased in the HFD group, compared with the chow control group. In particular, *CD36* mRNA expression in the liver was elevated by PPE feeding, while it was reduced by HFD relative to chow control (Figure 2A). Furthermore, the gene *CPT1* for fatty acid degradation was found to be expressed by PPE treatment, although without significance (*p* > 0.05), compared with the HFD control group (Figure 2D). These results indicated PPE significantly suppressed fatty synthesis in the liver which attenuated liver steatosis.

Abnormal secretion of cytokines (leptin, adiponectin, TNF-α, and IL-6) due to obesity may lead to insulin resistance in the body. Thus, the expression of insulin resistance-related genes in the liver was examined. The result showed that PI3K-Akt signaling pathway gene *IRS1*, *IRS2*, *PI3K*, and *AKT2* (Figure 2E–H) was up-regulated by PPE (medium-dose) supplementation. Moreover, a marked increase in mRNA expression of glycogen synthesis genes GSK-3β and GYS were observed in PPE_M treated rats (Figure 2I,J). In addition, we focused on the effect of dietary PPE on gluconeogenesis in the liver. Interestingly, the mRNA expression of biomarker genes *FOXO1*, *G6PC*, *FBP*, *and PEPCK* were significantly downregulated by PPE supplementation, especially in the high-dose group rats (Figure 2K–N). This revealed that the PPE supplementation was able to stimulate the PI3K-Akt signaling pathway and glycogen synthesis, and suppress gluconeogenesis in hepatic tissue.

The possible underlying mechanism of how PPE attenuates liver steatosis and insulin resistance in obese rats was illustrated in Figure 3. Taken together, these results confirmed that PPE supplementation is capable of regulating glucose and lipid metabolism to achieve metabolic benefits for obese rats.

### 3.4. PPE Attenuated Gut Microbiota Dysbiosis in Obese Rats and Maintained Glucose and Lipid Metabolism Homeostasis

PPE are bioactive peptides composed of 2–15 amino acids, some of which are absorbed and enter into the blood to reach various tissues and organs, while others travel through the digestive system to the guts, where it regulates the gut microbiota, which in turn affects body metabolism. The gut microbiota has evolved a sophisticated metabolic capacity to adapt to diverse dietary chemicals and environments. Therefore, the metagenomic sequencing of colon contents was performed to study how PPE supplementation impacted gut microbiome remodeling in obese rats. Compared with the Chow control group, HFD rats had substantially lower bacterial richness whereas PPE feeding enriched the phylogenetic diversity indicated by Chao1 (Figure 4A). Principal coordinate analysis (PCoA) (Figure 4B) and nonmetric multidimensional scaling (NMDS) (Appendix A) revealed distinct clustering of Chow control, HFD control, and PPE-supplementation groups, whereas a different dose of PPE groups was not clearly separated. Analysis of similarities (ANOSIM) analysis showed the difference between groups was significantly greater than that within groups (*p* = 0.001) (Appendix A). The community abundance bar plot on phylum level depicting the core microorganisms of rats was mainly composed of Firmicutes, Verrucomicrobia, Bacteroidetes, and Proteobacteria (Figure 4C). The results showed PPE treatment changed the ratio between different phylum gut microbiomes, especially in the high dose group; the ratio of Firmicutes/Bacteroidetes was significantly lower than that of the HFD group. Compared with the Chow control group, Firmicutes increased by 16% in the HFD group, whereas in the PPE_L/PPE_M/PPE_H groups it decreased (Appendix A). PPE feeding significantly enriched Verrucomicrobia and Bacteroidetes phylum in rats.

Compared with the Chow control group, the Verrucomicrobia decreased by 56% in the HFD group, however, in the PPE_H and PPE_L groups it increased by 218% and 136%, respectively (Appendix A). Bacteroidetes in the HFD group were 40% less than in the Chow control group, whereas the PPE_L/PPE_M/PPE_H intervention group increased by 50%, 175%, and 50% compared with that of the HFD group, respectively (Appendix A). Compared to Chow control rats, HFD rats displayed a significantly higher Firmicutes-to-Bacteroidetes ratio. Notably, PPE supplementation protected against this change to a certain extent. Proteobacteria in the HFD group decreased by 42% compared with that in the Chow control group, while the PPE_L/PPE_M intervention group increased to the same level as that in the Chow control group, and PPE_H increased by 191% compared with that in the HFD group (Appendix A).

We further performed linear discriminant analysis effect size (LEfSe) (LDA > 3) to identify taxa that may be microbiological markers for PPE_H, PPE_M, and PPE_L groups (Figure 5A, Appendix A). High-dose PPE intervention significantly elevated the abundance of Verrucomicrobia, Proteobacteria, *Akkermansia_muciniphila*, *Desulfovibrio_*sp., and *Romboutsia_ilealis*, and significantly reduced the abundance of Firmicutes. Moreover, medium-dose PPE intervention specifically enriched Muribaculaceae and Porphyromonadaceae of Bacteroidetes, Lactobacillales, Peptostreptococcaceae, Romboutsia, Faecalibacterium, Staphylococcaceae, Bacilli, *Romboutsia_ilealis*, *Clostridiales_bacterium_Marseille_P2846*, *Clostridium_*sp., *Anaerotruncus_*sp., *Lachnospiraceae_bacterium*, *Enterococcus_faecium* and *Lactobacillus_johnsonii*. Furthermore, pernicious bacteria were reduced, for instance, *Evtepia_gabavorous* and *Ruminococcus_gnavus*. Microbiological markers in the low-dose group were mostly similar to those in the medium-dose group, PPE supplementation at different doses significantly reduced the abundance of Firmicutes.

To elucidate the gut microbiome functional capacity changes impacted by PPE supplementation, functional annotation of the metagenome on the basis of the KEGG database was carried out. Our metagenomic analysis indicated that after PPE supplementation, the metabolic pathways in glucolipid-related metabolism were significantly altered in obese rats (Figure 5B), which has demonstrated the efficacy of dietary PPE. The most remarkably altered pathways were those involved in carbohydrate metabolism (Glycolysis/Gluconeogenesis, ko 00010), lipid metabolism (Glycerolipid metabolism, ko 00561 and Sphingolipid metabolism, ko 00600), global and overview maps (Carbon metabolism, ko 01200 and Fatty acid metabolism, ko 01212), amino acid metabolism (Alanine, aspartate and glutamate metabolism, ko 00250), membrane transport (ABC transporters, ko 02010), xenobiotics biodegradation and metabolism (Drug metabolism—other enzymes, ko 00983), biosynthesis of other secondary metabolites (Streptomycin biosynthesis, ko 00521) and antimicrobial (Cationic antimicrobial peptide resistance, ko 01503).

### 3.5. PPE Promotes SCFAs Production Especially Butyric Acid

We further investigated SCFAs concentrations in rat feces using GC, and found that the dominant SCFAs were acetic acid, propionic acid, and butyric acid. They were significantly (*p* < 0.05) increased in the PPE_M group (Figure 6). The contents of isobutyric acid, isovaleric acid, and valeric acid were minor in all the groups and slightly (*p* < 0.05) elevated by PPE supplementation. Overall, compared with the HFD Control group, the PPE supplementation groups prominently increased the content of total SCFAs.

### 3.6. Impact of PPE on the Metabolic Profile in Blood Serum

The serum metabolome contains a plethora of biomarkers and causative agents of various diseases, while the metabolites are influenced by both the gut microbiome and lifestyle choices such as smoking, alcohol, or diet [40]. To further explore the effects of PPE supplements on the blood serum metabolic response in obese rats, we employed a non-targeted metabolomics technique to characterize the serum metabolome. According to the peak heights in the spectra, about 4052 peaks of positive ions (ESI+) and 4771 peaks of negative ions (ESI−) were detected, which were extracted from the total ion chromatograms by MassLynx. By database comparison, 726 metabolites were identified in serum samples. The list of all the metabolites was shown in Appendix A, including Metabolite ID, *m*/*z*, mode, retention time, and the abundance of metabolites in each sample. These metabolites can be further divided into ten classes (reference Human Metabolome Database 4.0), with 399 (69.15%) lipids and lipid-like molecules, 47 (8.15%) organic heterocyclic compounds, 39 (6.76%) organic acids and derivatives, 26 (4.51%) organic oxygen compounds, 25 (4.33%) benzenoids, 16 (2.77%) organic nitrogen compounds, 15 (2.60%) phenylpropanoids and polyketides, 5 (0.87%) nucleosides, nucleotides, and analogues, 3 (0.52%) hydrocarbons and 2 (0.35%) alkaloids and derivatives (Figure 7B). The partial least squares discriminant analysis (PLS-DA) showed that the metabolic profiles of PPE intervention rats were significantly different from that of untreated groups, while the metabolic profiles of high-fat diet versus chow diet fed rats were clearly distinguishable. However, different dose of PPE intervention rats was not completely separated from each other (Figure 7A). To maximize the discrimination between PPE-treated and untreated obesity rats, we next performed pairwise OPLS-DA on the serum metabolome data from these groups. As revealed by the volcano plots, substantial metabolic changes occurred after PPE supplementation in obese rats (Figure 7C).

Among the significantly changed molecules, the top 30 differential metabolites were selected as potential biomarkers (VIP value > 1) among these groups (Figure 8A and Appendix A). As shown in the heatmap, the glycerolipids, sterol lipids, prenol lipids, and carbohydrates, and carbohydrate conjugates (maltotriose and manninotriose) were remarkably increased, in comparison to the chow control rats. Besides, glycerophospholipids were decreased in high-fat-diet obese rats’ serum. Furthermore, the PPE-treated group (medium-dose) was found to down-regulate fatty acyls, prenol lipids, and glycerophospholipids metabolites while it up-regulated a variety of metabolites such as lactones, indoles and derivatives, sphingolipids, glycerophospholipids, carboxylic acids and derivatives, tetrapyrroles and derivatives, steroids and steroid derivatives, tetrahydroisoquinolines and pyridines and derivatives metabolites.

To achieve a deeper understanding of the differential serum metabolites changes in response to high-fat diet and PPE supplementation, pathway enrichment analysis was performed with MetaboAnalyst annotated with KEGG (Figure 8B and Appendix A). The results revealed significant changes in lipid metabolism, such as regulation of lipolysis in adipocytes, glycerophospholipid metabolism, bile secretion, protein digestion and absorption, and hypotensive-related metabolic pathways of renin secretion, which was in accordance with the KEGG analysis of intestinal microbiome metagenomic data.

## 4. Discussion

In this study, PPE supplementation was found to drastically ameliorate diet-induced obesity through modulating organ metabolism and reconfiguring gut microbiota. It is found that the PPE supplementation enabled reduction of body weight and ameliorated serum glucose and lipid indexes, inhibited liver lesions, and enhanced protective immunity. Furthermore, PPE supply could attenuate liver steatosis and insulin resistance by regulating key gene expression. PPE was observed to rehabilitate gut microbiota dysbiosis, to promote SCFAs production, and to maintain glucose and lipid metabolism homeostasis. Overall, the medium-dose PPE treatment displayed a pronounced therapeutic effect through the interwoven interactions between gut, liver, and blood circulation.

Obesity/overweight continues its relentless global rise, with the number of people with excess body weight reaching >2 billion, about 30% of the world population [41]. As a result, this type of chronic disorder causes high morbidity and mortality [42]. Although *Spirulina platensis* protein hydrolysate and C-phycocyanin has been reported to possess hypolipidemic and hypocholesterolemic effects in high-fat diet-fed mice [12,21,30,43], few studies have investigated the possible therapeutic benefits of Spirulina platensis protein hydrolysate or C-phycocyanin on impaired glucose-lipid homeostasis. Here, we demonstrated that PPE supplementation was able to attenuate gut microbiota dysbiosis and to restore impaired glucose and lipid metabolism homeostasis in obese rats.

In our previous publication [28], we reported that PPE was rich in bioactive peptides, which have effective antioxidant, hypotensive, hypoglycemic, and immunomodulatory activities in vitro. In this study, we further investigated the active role of PPE in vivo in obesity model rats with a long-term high-fat diet. PPE supplementation was seen to significantly reduce body weight, ameliorate glucose clearance ability and serum lipid index, and decrease serum leptin levels, while the regulation of serum insulin and adiponectin level was not obvious. Our results indicated that the PPE treatment of the PPE_H group was effective in the reduction of serum leptin level, while it was not better than the PPE_M group in the reduction of body weight. Leptin is the hormone secreted by adipose tissue in direct proportion to its mass, which is able to increase energy release [44]. Thus, the reduction in leptin levels of the PPE_H group was associated with a decrease in adipose tissue weight. We postulated that for the PPE_H group the weight loss was mainly attributed to the reduction of fat weight, while a high protein diet could increase the rats’ skeletal muscle weight which might have made the overall body weight of rats in the PPE_H group to be higher than PPE_M group. Furthermore, the PPE invention indicated an obvious therapeutic effect on abnormal fatty liver and a powerful anti-inflammatory ability. The results revealed that although the high-fat diet caused a glucose and lipid metabolism disorder, PPE possessed a positive effect on serum glucose-lipid profile levels and liver steatosis.

The liver is a primary tissue to accommodate glucose and lipid metabolism and plays a crucial role in regulating the energy metabolic balance, and maintaining glucolipid homeostasis. It is also responsible for nutritional transport and immune response [45]. Firstly, we explored the genetic regulation of fat entering the liver by investigating how a scavenger receptor *CD36* transfers fatty acids from serum to liver cells [46]. It was observed that *CD36* mRNA expression in the liver was elevated by PPE feeding, which illustrated more fatty acids were directed into the liver cells. However, the fatty acid degradation gene CPT1 was seen to increase by PPE treatment (with no significant difference). Interestingly, we found that fatty acid biosynthesis gene *ACC* and *FASN* were markedly (*p* < 0.001) reduced by PPE feeding. By stark contrast, they were significantly (*p* < 0.001) increased in the HFD group, compared with the chow control group. Therefore, PPE was believed to effectively inhibit the synthesis of fat in the liver, so as to avoid the appearance of fatty liver.

Moreover, Obesity causes a reduction of glycogen production and gluconeogenic gene expression in the liver [47]. Interestingly, the rats administrated with PPE would upregulate glycogen synthesis genes *GSK-3* and *GYS*, while gluconeogenic biomarker genes *FOXO1*, *G6PC*, *FBP*, *and PEPCK* were significantly downregulated. Notably, PI3K-Akt signaling pathway gene *IRS1*, *IRS2*, *PI3K*, *and AKT2* was upregulated by PPE supplementation. This demonstrated that PPE would inhibit TNF signaling pathway, stimulate the PI3K-Akt signaling pathway, promote the synthesis of glycogen, and suppress gluconeogenesis. Therefore, it could reduce the serum glucose concentration to achieve the preferred homeostasis of blood glucose level. A similar mechanism of dietary supplementation with sea-cucumber-derived ceramides and glucosylceramides alleviating insulin resistance in high-fructose-diet-fed rats was reported in recent research [48].

Current study shows that obesity has a tight linkage with the gut microbiome and host metabolism [4,6,49]. As an important modulator of the gut microbiota, the diet was found to be able to modulate the composition and function of this community of microbes in humans and other mammals [50,51]. Therefore, dietary intervention became a useful approach to improving metabolic syndromes in association with the alteration of gut microbiota in high-fat diet-fed rats [8]. In this study, we have proven that PPE supplementation was capable of enhancing the diversity and composition of gut microbiota. Previous studies have shown that there exists a higher abundance of Firmicutes, as well as a higher Firmicutes/Bacteroidetes ratio for obese patients in comparison to normal people. Our work indicated that after dietary treatment, there was an increase in Bacteroidetes and a decrease in Firmicutes for obese patients [3]. It was found that Firmicutes was increased in the HFD group, whereas it was significantly decreased in PPE_L/PPE_M/PPE_H groups, in comparison to the Chow control group. On the other hand, Bacteroidetes was decreased in the HFD group, whereas it was significantly increased in the PPE_L/PPE_M/PPE_H intervention group. Although HFD rats displayed a significantly higher Firmicutes-to-Bacteroidetes ratio, compared to Chow control rats, PPE supplementation was seen to decrease this ratio to a certain extent. Notably, an abundance of probiotic *Akkermansia_muciniphila* was significantly increased in all the PPE supplementation groups. There was clear evidence that an increase in the abundance of the genus *Akkermansia* resulted in the alleviation of obesity. It is also found that *A. muciniphila* supplementation may improve insulin sensitivity and blood lipid index, decrease body weight and fat mass and reverse liver dysfunction and inflammation biomarkers [52]. Moreover, PPE supplementation was believed to elevate the abundance of *Verrucomicrobia*, *Proteobacteria*, *Lactobacillales*, *Peptostreptococcaceae*, *Romboutsia*, *Faecalibacterium*, *Staphylococcaceae*, *Clostridiales_bacterium*, *Clostridium_*sp, *Anaerotruncus_*sp., *Lachnospiraceae_bacterium*, *Enterococcus_faecium*, *Lactobacillus_johnsonii,* and *Desulfovibrio_*sp., whereas pernicious bacteria were reduced, for instance, *Evtepia_gabavorous* and *Ruminococcus_gnavus*. *Proteobacteria* and *Verrucomicrobia* were reported to have a higher abundance in healthy people [6]. Lactobacillus is a kind of probiotic bacteria that exists in the human body, which has an inhibitory effect on the proliferation of pathogenic bacteria, and could stimulate mucin secretion, and relieve inflammatory response and obesity [53]. *Romboutsia genus*, which belongs to the Peptostreptococcaceae family, plays a key role in maintaining host health which makes this taxon a valuable intestinal biomarker. It also has the ability to produce SCFAs, especially butyric acid [54]. *Faecalibacterium* genus was reported to enhance butyrate formation [55] and *Faecalibacterium prausnitzii* is well known as a functionally important member of the microbiota and has an impact on host physiology and health [56]. PPE elevated SCFA-producing *Lachnospiraceae_bacterium* [57] species in the intestine. *Clostridiales_bacterium* was shown to be a butyrate-producing bacterium that could regulate the gut microbiome of type 2 diabetes, and reduce the production of harmful compounds such as indoles and hydrogen sulfide [58]. *Clostridium* sp. isolated from feces has the ability to degrade cellulose, and produce small molecular sugars, such as glucose, which can be easily absorbed and utilized. This bacterium is able to promote the reproduction of beneficial bacteria [59]. *Enterococcus faecium* which belongs to gram-positive bacteria is a human probiotic, which assists the intestinal tract to digest food and protect gastrointestinal mucosa. It can also improve the intestinal microenvironment to maintain the balance of the gut microbiome [60]. A study in healthy dogs reported that oral application of *Enterococcus faecium* strain EE3 led to decreased total blood lipid concentration and also brought cholesterol concentrations back to the reference range [61]. *Lactobacillus_johnsonii* could ameliorate diet-induced obesity and hyperlipidemia [62]. It has a significant effect on the reduction of body weight gain, on the plasma lipid level in the liver and white adipose tissue, and on the improvement of insulin resistance [62]. Unfortunately, PPE could not reverse the HFD-induced elevation in *Staphylococcaceae* and *Desulfovibrio_sp*, which is a pathogenic microbe in the gut. Despite this, PPE supplementation could remarkably decrease the pernicious bacteria. *Evtepia_gabavorous* can lead to anxiety, depression, and restlessness by reducing the neurotransmitter GABA (γ-aminobutyric acid) [63]. *Ruminococcus_gnavus* is considered to be an important part of the altered microbiome in inflammatory bowel disease (IBD), IBD is usually accompanied by an increase in *Ruminococcus_gnavus* [64]. To sum up, these data verified that PPE supplementation in obese rats had remarkably altered the gut microbiota community structure and attenuated gut microbiota dysbiosis induced by diet. Overall, our metagenomic analysis focused on the functional genome of gut microbiota in obese rats after PPE supplementation, it revealed that dietary PPE was capable of altering glucolipid-related metabolic pathways for improvement in human health.

Fermentable carbohydrates might alter cecal pH value and SCFAs production, which could assist in maintaining intestinal function and health [65]. We also explored how the PPE administration would promote SCFA production, mainly as acetic, propionic, and butyric acid. This may be attributed to increased SCFA-producing taxa *Romboutsia genus*, *Faecalibacterium genus*, *Lachnospiraceae_bacterium*, *and Clostridiales_bacterium* that have been mentioned in the previous paragraph. SCFAs also affect lipid metabolism [62]. Acetic acid is the most abundant SCFA in the intestine and could downregulate the expression of the fatty acid synthase gene [66]. It has been demonstrated that propionic acid may inhibit the synthesis of cholesterol in the liver [67]. Consistently, the TC and TG levels in the PPE supplementation group were remarkably lower compared with HFD group rats. Butyrate is used as a fuel for cells since it could improve insulin sensitivity and increase energy expenditure by induction of mitochondria function [68]. Overall, PPE supplementation was found to enhance SCFA production through altering the gut microbiome, which has the effect of improvement of glucose and lipid metabolism.

It is now widely accepted that the gut microbiome may significantly shape the utilization of metabolic pathways in the host, and changes in gut microbiota may affect the host’s circulating metabolites [69]. Non-targeted metabolomics data illustrated that daily oral supplementation of PPE was able to alleviate diet-induced obesity. It was found that 69.15% of differential metabolites in the metabolome data were lipids and lipid-like molecules, which indicated a remarkable alteration in lipid metabolism. The metabolic pathway enrichment analysis of differential metabolites was performed to explore the active metabolic pathways related to obesity. The results revealed that there were significant changes in lipid metabolism. It has been reported in the literature that algae extracts could modulate the gut microbiome and metabolic diseases such as diabetes [8,70], hepatic metabolism disorder [71], and hypertension [23]. These studies have shown that algae extracts were a potential therapeutic dietary supplement for metabolic disease.

Based on the research outputs from biochemistry, cell physiology and metabolism, and bioinformatic analysis of gut microbiota and serum metabolites discussed above, we established a metabolic model to describe how the synergy of PPE functions would enable cell metabolic reprogramming to prevent the aggravation of obesity. This model elucidated the underlying mechanism of PPE treatment as: 1. PPE inhibits the fatty synthesis pathway to attenuate liver steatosis. 2. PPE attenuates insulin resistance by down-regulating TNF signaling pathway, activating the PI3K-Akt signaling pathway to suppress glycogen synthesis and gluconeogenesis. 3. PPE reconfigures gut microbiota and promotes SCFA production. 4. PPE modulates lipid metabolism in blood circulation.

## 5. Conclusions

In summary, our work has proven the hypothesis that the bioactive peptides extracted from algae is a novel and sustainable dietary source with substantial potential to be an efficient therapeutic dietary supplement for patients with obesity and other metabolic disorders. We have applied multi-omic analyses to validate our hypothesis that PPE supplementation is able to alleviate diet-induced obesity by reconfiguration of gut microbiota and circulating metabolites. The study offered new insights into the correlation between the gut microbiota and circulating metabolites as a suitable target for the treatment of metabolic disorders. For the first time, it presented a model for a possible underlying mechanism with which PPE may reprogram the gut microbiota and serum metabolism for an efficient therapeutic dietary treatment for obese populations. This study thus provides a theoretical basis for the continued utilization of bioactive peptides derived from algae.

## Figures and Tables

**Figure 1 nutrients-14-03635-f001:**
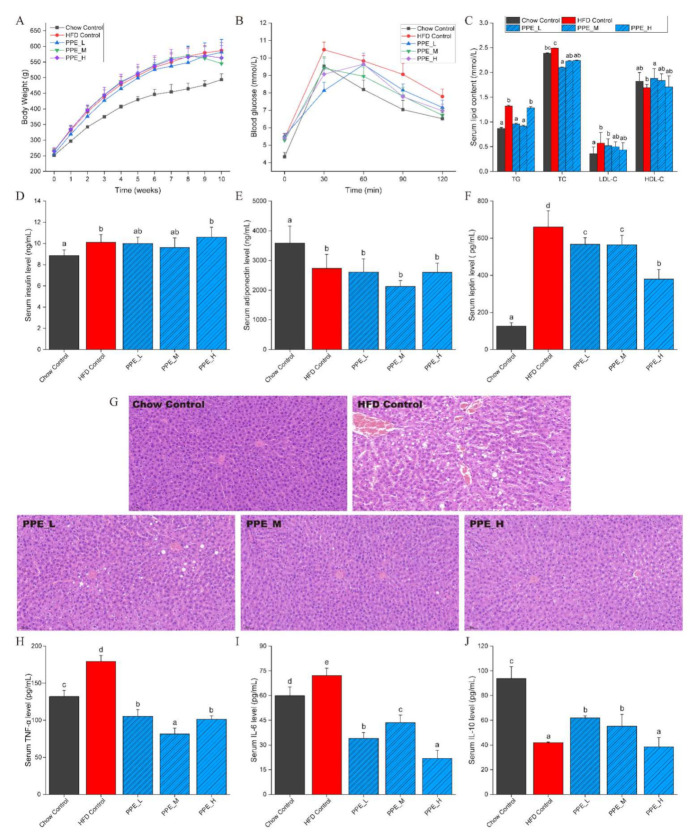
Effects of PPE on obesity, glucose homeostasis, hepatic injury, and inflammatory cytokines level in serum. (**A**) Body weight; (**B**) Blood glucose response curve during the insulin tolerance test; (**C**) TC, TG, LDL-C, and HDL-C contents in serum; (**D**) Serum insulin level; (**E**) Serum adiponectin level; (**F**) Serum leptin level; (**G**) Histological assessment of liver by H&E staining (400× magnification, 50 μm); (**H**) Serum TNF-α level; (**I**) Serum IL-6 level; (**J**) Serum IL-10 level. Data are presented as mean ± SD (*n* = 6 rats per group). Data with different superscript letters (a–d) are significantly different (*p* < 0.05) between each group using one-way ANOVA followed by Tukey’s multiple comparison post-test. PPE: phycobiliproteins bioactive peptides extract; HFD: high-fat diet group; PPE_L: low-dose PPE treatment group; PPE_M: medium-dose PPE treatment group; PPE_H: high-dose PPE treatment group; TG: triglyceride; TC: total cholesterol; LDL-C: low-density lipoprotein cholesterol; HDL-C: high-density lipoprotein cholesterol; TNF-α: tumor necrosis factor-α; IL-6: interleukin 6; IL-10: interleukin 10.

**Figure 2 nutrients-14-03635-f002:**
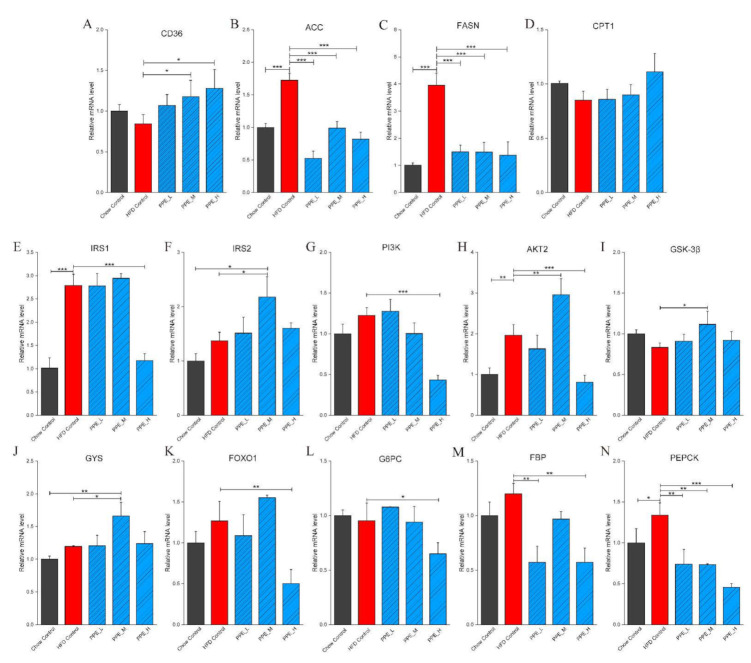
Effects of PPE supplementation on hepatic carbohydrate and fatty acid metabolism in obese rats. (**A**–**D**) The mRNA expression related to fatty acid transfer, synthesis and β-oxidation. (**E**–**N**) The mRNA expression related to insulin signaling, glycogen synthesis and gluconeogenesis in the liver. (* represents *p* < 0.05, ** represents *p* < 0.01, *** represents *p* < 0.001, *n* = 6 rats in each group) PPE: phycobiliproteins bioactive peptides extract; HFD: high-fat diet group; PPE_L: low-dose PPE treatment group; PPE_M: medium-dose PPE treatment group; PPE_H: high-dose PPE treatment group.

**Figure 3 nutrients-14-03635-f003:**
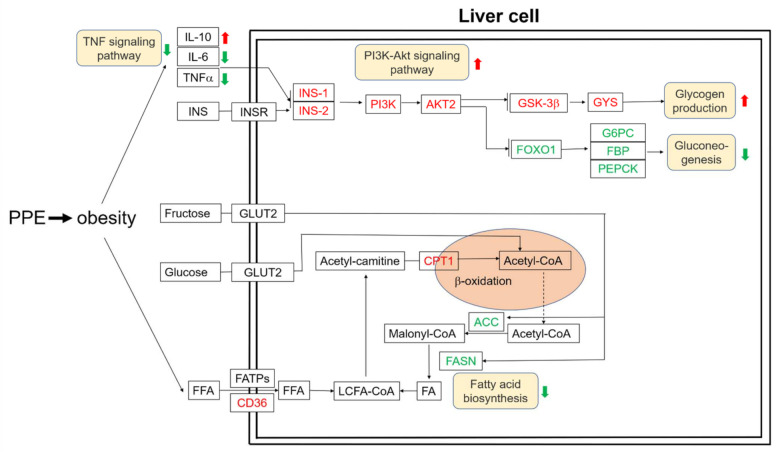
The possible underlying mechanism of how PPE attenuates liver steatosis and insulin resistance in obese rats’ liver. Gene name with red color indicates up-regulated while green color indicates down-regulated. Red color arrows represents up-regulated while green color arrows represents down-regulated.

**Figure 4 nutrients-14-03635-f004:**
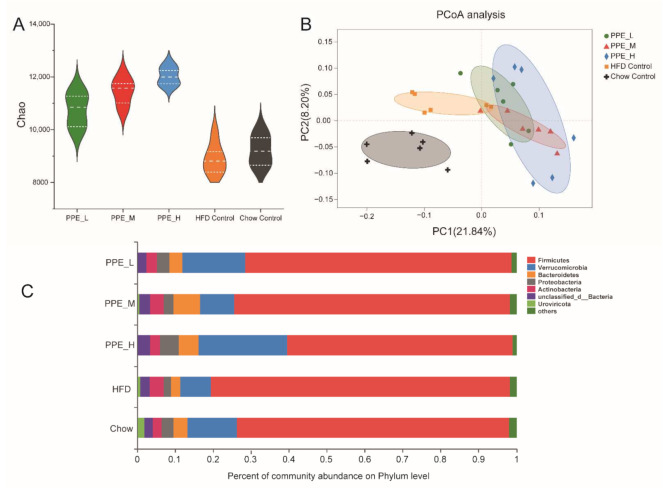
PPE alters gut microbiota composition in obese rats. (**A**) Alpha diversity indexes Chao of gut microbiota at the genus level in the colonic contents of each group of rats. (**B**) PCoA of the gut microbiome composition at the genus level based on the Binary-jaccard distance for five groups (PPE_L, PPE_M, PPE_H, HFD Control, Chow Control) rats (*n* = 6). (**C**) The community abundance bar plot on the phylum level of five groups (PPE_L, PPE_M, PPE_H, HFD Control, Chow Control) rats. (*n* = 6 rats in each group) PPE: phycobiliproteins bioactive peptides extract; HFD: high-fat diet group; PPE_L: low-dose PPE treatment group; PPE_M: medium-dose PPE treatment group; PPE_H: high-dose PPE treatment group; PCoA: Principal coordinate analysis.

**Figure 5 nutrients-14-03635-f005:**
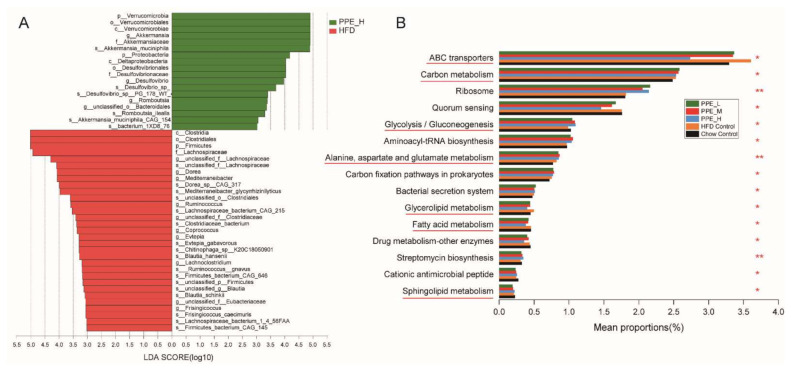
The biomarker gut bacteria in high-dose PPE supplement obese rats and its function. (**A**) Differentially expressed bacteria obtained from LEfSe analysis (LDA > 3) of gut microbiota in obesity rats fed with PPE_H. (**B**) Summary of the top 15 distinct metabolic pathways based on metagenomic data of colonic contents samples from five groups of rats fed with different diets. Kruskal-Wallis H test, * 0.01 < *p* ≤ 0.05, ** 0.001< *p* ≤ 0.01. (*n* = 6 rats in each group) PPE: phycobiliproteins bioactive peptides extract; HFD: high-fat diet group; PPE_L: low-dose PPE treatment group; PPE_M: medium-dose PPE treatment group; PPE_H: high-dose PPE treatment group; LDA: Linear discriminative analysis.

**Figure 6 nutrients-14-03635-f006:**
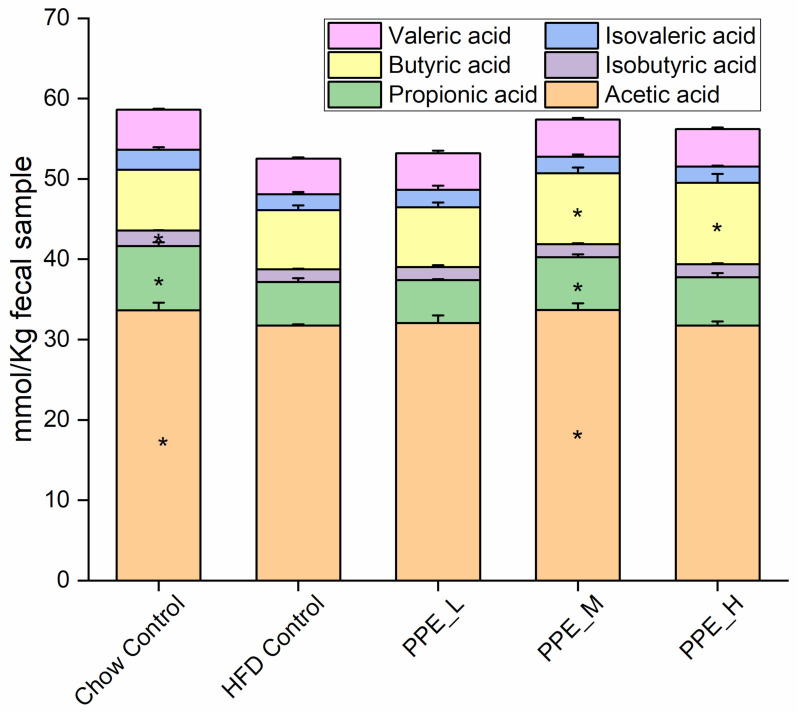
PPE promote SCFA production (*n* = 6 per group). Individual SCFA levels in fecal content. * significantly different (*p* < 0.05), compared with the HFD Control group. Data are the mean ± standard error of the mean.

**Figure 7 nutrients-14-03635-f007:**
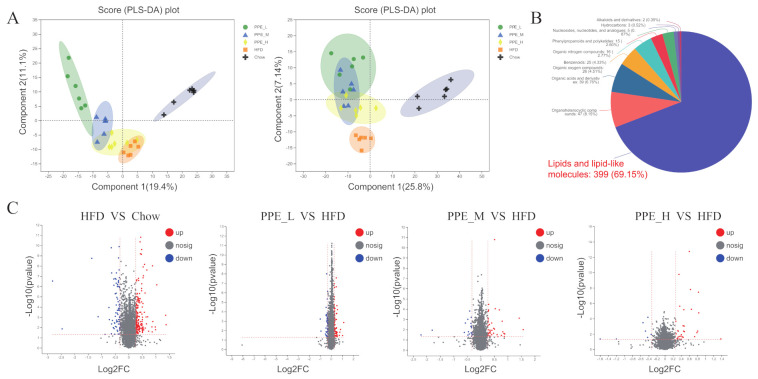
Effects of PPE on serum metabolomic profiling by UHPLC-Q Exactive MS. (**A**) PLS-DA score plot of the five groups (PPE_L, PPE_M, PPE_H, HFD Control, Chow Control) in the positive ion (left) and negative ion (right). (**B**) Pie chart of metabolite classification information compared with HMDB 4.0 database. (**C**) Volcano plot (based on the combination of positive and negative ions) between HFD & Chow, PPE_L & HFD, PPE_M & HFD, and PPE_H & HFD groups. (*n* = 6 rats in each group) PPE: phycobiliproteins bioactive peptides extract; HFD: high-fat diet group; PPE_L: low-dose PPE treatment group; PPE_M: medium-dose PPE treatment group; PPE_H: high-dose PPE treatment group; PLS-DA: Partial least squares discriminant analysis.

**Figure 8 nutrients-14-03635-f008:**
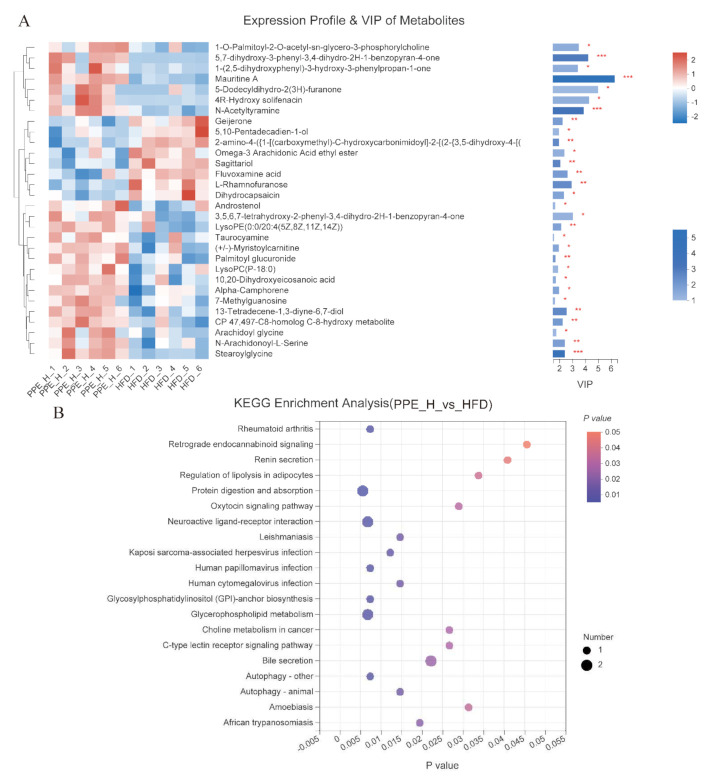
The different metabolites in high-dose PPE supplement obese rats and KEGG enrichment pathways. (**A**) Expression profile and VIP score (OPLS-DA) of significantly different metabolites (left, top 30 are shown, * represents *p* < 0.05, ** represents *p* < 0.01, *** represents *p* < 0.001). (**B**) KEGG enrichment analysis (right, top 30 are shown) between PPE_H diet & HFD diet. (*n* = 6 rats in each group) PPE: phycobiliproteins bioactive peptides extract; HFD: high-fat diet group; PPE_L: low-dose PPE treatment group; PPE_M: medium-dose PPE treatment group; PPE_H: high-dose PPE treatment group; OPLS-DA: orthogonal partial least squares discriminate analysis; VIP: variable importance in the projection.

## Data Availability

All data generated or analyzed during this study are included in this published article [and its Appendix A].

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
