# Peer review of "Reconfiguration of Gut Microbiota and Reprogramming of Liver Metabolism with Phycobiliproteins Bioactive Peptides to Rehabilitate Obese Rats"

_nutrients, 2022, doi:10.3390/nu14173635_

Round 1

Reviewer 1 Report

Dear Authors

The manuscript is interesting and deserves consideration in this context.

Since there is a huge work of characterization beyond these big amount of data, I ask the authors to add extraction yields and peptides MS/HPLC data for their identification. Consider also to add some more recent literature regrading bioactive peptides from natural sources, such as: "Natural Occurring β-Peptides: A Fascinating World of Bioactive Molecules","Plant-derived peptides rubiscolin-6, soymorphin-6 and their c-terminal
amide derivatives: Pharmacokinetic properties and biological activity".

Overall my recommendation is minor revision after changes.

Author Response

We are very grateful for the constructive and insightful comments from the anonymous reviewers. Our point-to-point responses to their comments has been submitted as a Word file. Please see the attachment.

Sincerely yours,

Pengcheng Fu

Reviewer 2 Report

In order to unveil the role of phycobiliproteins bioactive peptides extracts (PPE) in treatment of diet-induced obesity, the authors investigated the systemic changes of levels of genes related lipid metabolites in liver, gut microbiota, and metabolites in serum and feces from high fat-induced obese rats treated with different doses (low, medium and high dose) of PPE. However, before this manuscript is ready for publication, some concerns need addressing.

  1. Why did the authors emphasize observed changes were from cell metabolism in the title?
  2. How did the authors define the dose of PPE used in high fat diet groups? Provided in this study, doses of low, medium and high PPE were very close. Please provide more information on the dose of PPE described in the manuscript.
  3. In Figure 1, the authors investigated weight loss of rats, blood glucose, inflammatory cytokines levels in serum. However, most of levels did not respond to treatment of PPE in a dose dependent manner. Please clarify it.
  4. The authors mentioned words of “metabolomic” and “metabonomic”. What was the difference between “metabolomic” and “metabonomic”?
  5. In Figure 7, the authors ran the serum samples for global profiling at both modes, including positive and negative modes. Why did the changed metabolites look completely different at different dose of PPE treatment?
  6. How did the author identify the metabolites shown in of Figure 7D? How did the authors verify them?
  7. The authors investigated changes of gut microbiota and metabolites led by treatment of PPE. Had the authors further studied correlations between gut microbiota and metabolites observed in the results?

Author Response

We are very grateful for the constructive and insightful comments from the anonymous reviewers. Our point-to-point responses to the comments has been submitted as a Word file. Please see the attachment.

Sincerely yours,

Pengcheng Fu

Round 2

Reviewer 2 Report

Thanks for responses from the authors. However, some comments were not replied reasonably. All in all, the authors did a lot of work. However, the data was not fully explained/understood. The manuscript needed more improvements.

1.       Point 2: How did the authors define the dose of PPE used in high fat diet groups? Provided in this study, doses of low, medium and high PPE were very close. Please provide more information on the dose of PPE described in the manuscript.

The explanation did not support the design of doses applied in the manuscript. There should be some guidance how to design doses of low, medium and high.

2.       Point 3: In Figure 1, the authors investigated weight loss of rats, blood glucose, inflammatory cytokines levels in serum. However, most of levels did not respond to treatment of PPE in a dose dependent manner. Please clarify it.

Described by the authors, high dose and medium dose led to inconsistent results. The explanations provided by the authors were not convincing.  

3.       Point 6: How did the author identify the metabolites shown in of Figure 7D? How did the authors verify them?

The authors did not answer how to identify changed metabolites induced by treatment of PPE. Compared with public database such as HMDB, identification of metabolites was not acceptable. The list of these metabolites should be provided, including m/z , retention time, MS/MS and so on.

Author Response

Dear Reviewer:

We have updated our response and explanation, please see the attachment.

Thank you very much for your kind consideration.

Sincerely yours,

Pengcheng Fu
